# Venous Thromboembolism Prophylaxis and Thrombotic Risk Stratification in the Varicose Veins Surgery—Prospective Observational Study

**DOI:** 10.3390/jcm9123970

**Published:** 2020-12-07

**Authors:** Krzysztof Wołkowski, Maciej Wołkowski, Tomasz Urbanek

**Affiliations:** 1Department of Surgery, Saint Ann’s Hospital, 32-200 Miechów, Poland; k.wolkowski@poczta.onet.pl; 2Department of General Surgery, Vascular Surgery, Angiology and Phlebology, Medical University of Silesia, 40-750 Katowice, Poland; maciek2106a@gmail.com

**Keywords:** varicose veins, surgery, deep vein thrombosis, prophylaxis, Caprini score

## Abstract

Background: An invasive phlebological treatment is still not free from complications such as thrombosis. As in other surgical populations, not only the treatment modality, but also patient condition-related venous thromboembolism (VTE) risk factors matter. The current protocols used in varicose vein surgery centers are based mostly on individual risk assessment as well as on an implementation and extrapolation of general surgery VTE prophylaxis guidelines. In the presented study, the efficacy of routine VTE pharmacological thromboprophylaxis in patients undergoing saphenous varicose vein surgery was prospectively evaluated. In the result assessment, VTE risk factor evaluation and Caprini score results were included; however, due to the limited size of the projected study group, as well as expected limited deep vein thrombosis (DVT) prevalence in this clinical scenario, it was not possible to perform the validation of the Caprini model efficacy in the projected study model. Methods: In the study, 141 patients undergoing saphenous vein stripping and miniphlebectomy in spinal anesthesia were included. In all of the patients, VTE risk factors (including Caprini score evaluation) were assessed, and the routine thromboprophylaxis with enoxaparin 40 mg for 10 days was used. The venous ultrasounds were undertaken before the surgery and on the 10th and 30th day after surgery. The study endpoint was the presence of symptomatic or asymptomatic DVT confirmed in the imaging study. The study safety endpoint was major bleeding occurrence intraoperatively or within 30 days after surgery. Results: The presence of a postoperative DVT was diagnosed in five cases (3.5%) In all of these cases, only distal DVT was confirmed. Despite extensive saphenous varicose vein surgery with stripping and miniphlebectomy performed in nontumescent but spinal anesthesia, no proximal lower leg episode was diagnosed. Three out of five DVT cases were diagnosed on day 10 postoperative control, while a further two were confirmed in the ultrasound examination performed 30 days after procedure. No clinically documented pulmonaly embolism (PE) as well as no bleeding episodes were noticed. Among the factors related to the statistically significant higher DVT occurrence, the results of the Caprini score were identified with odds ratio (OR) = 2.04 (95% CI = (0.998; 4.18)). Another factor that became statistically significant in terms of the higher postoperative DVT prevalence was the reported Venous Clinical Severity Score (VCSS) results (OR = 1.98; 95% CI (1.19; 3.26)). In the multiple logistic regression analysis, the patient age (OR = 0.86; 95% CI (0.75–0.99)), Caprini score evaluation results (OR = 4.04; 95% CI (1.26–12.9)) and VCSS results (OR = 2.4; 95% CI (1.23–4.7)) were of statistical significance as predictors for postoperative DVT occurrence, with a *p* value of 0.029 for age, and *p* = 0.017 and *p* = 0.009 for Caprini score results and VCSS results, respectively. Due to the confirmed limited number of the DVT events in our study cohort, as well as the descriptive and explorative nature of the achieved results, the final clinical potential and significance of the identified parameters, including Caprini score rate and VCSS rate, should be interpreted with caution and studied in the further trials in these clinical settings. Conclusion: All the patients undergoing varicose vein surgery should undergo VTE risk evaluation based on the individual assessment. In VTE risk evaluation, patient and surgical procedure characteristics based on the factors included into the Caprini score but also on specific chronic venous disease-related factors should be taken into consideration. Further studies are needed to propose an objective and validated VTE risk assessment model, as well as a validated antithrombotic prophylaxis protocol in this particular patient group.

## 1. Introduction

Technical progress, experience and enhanced knowledge have resulted in significant changes in varicose vein treatment. Minimal invasive treatment modalities and thermal or nonthermal ablation methods have replaced open surgery in many centers of phlebology [1,2]. Significant progress has also been noticed in the traditional surgical therapy of varicose veins. Clinically important changes include rapid postoperative patient mobilization, the use of tumescent anesthesia, ultrasound and minimal cosmetic incisions [3].

Despite this progress, invasive phlebological treatment is still not free from complications such as thrombosis. If assessed on the basis of the retrospective studies and evaluation, the prevalence of venous thromboembolism in the population of the patients undergoing varicose vein surgery seems to be relatively low with the rate of the clinically symptomatic deep vein thrombosis (DVT) usually not exceeding 1% [4]. Hagmuller et al., in a retrospective study on 3300 varicose veins surgery patients, documented the presence of symptomatic DVT in 0.15% and clinically symptomatic pulmonary embolism (PE) in 0.06% of all cases, while Critchley, in a group of 973 varicose vein surgery patients, found that DVT cases were represented by a rate of 0.5% [5]. In other retrospective studies, a symptomatic DVT rate of from 0.4 to 0.7% was documented [6,7,8]. 

Contrary to the available retrospective reports, in prospective observational studies, a much higher DVT rate was documented, especially when the objective imagine method was used. Van Rij, in a group of 377 patients undergoing saphenous vein stripping and phlebectomy, confirmed a 5.3% prevalence of DVT. In most of the cases, the distal (90%) and asymptomatic DVT was diagnosed on the basis of venous ultrasound examination [9]. An almost similar DVT rate (4.8%) was documented in the prospective study performed by Puttaswamy and coworkers [10]. 

Thrombotic complications are not limited to surgical patients. Despite the minimal invasive character of the currently proposed phlebological invasive treatment, after this kind of therapy, venous thromboembolism (VTE) cases also occur. According to the reported data concerning endovenous thermal ablation, the rate of deep vein thrombosis does not exceed usually 1%. The use of the minimal invasive approach based on the endovenous saphenous ablation can also result in the special, method-related complications, such as EHIT (endovenous heat-induced thrombosis) [1,4]. 

As in other surgical and medical populations, not only the treatment modality, but also the patient characteristics and patient condition-related VTE risk factors matter. According to current knowledge, as well as previously performed studies, several VTE risk factors can be identified in most patient populations undergoing invasive treatment (including patient age, obesity, hormonal treatment, previous DVT and others) [4,11,12]. When calculating the risk of DVT development related to the particular treatment, both, procedure- and patient-related factors should be taken into consideration [4,11,12]. 

One of the most difficult-to-solve problems regarding proper VTE thromboprophylaxis application seems to be proper VTE risk assessment [4,12]. Among some of the proposed models, the Caprini score has been put forward and has nowadays been implemented in many centers, in many patient populations, as well as in many clinical situations [13,14]. The clinical utility of the Caprini score and its efficacy have been validated in general surgery, otolaryngology, plastic surgery, intensive care and other clinical settings [14,15,16,17]. However, to date, dedicated studies focusing on Caprini score validation in varicose vein open surgery are still lacking. Among the available studies in the field of the invasive varicose vein treatment that include the Caprini score, the study of Rhee and coworkers should be mentioned [18]. In this study, we focused on the EHIT prevalence in patients undergoing endovenous laser ablation. Among the factors influencing higher EHIT rate, previous DVT episodes, CEAP classification C3–C6, male gender and a high Caprini score were identified [18]. 

Beyond proper VTE risk assessment in varicose vein patients, the proper antithrombotic prophylaxis algorithm should also be followed. The current protocols used in varicose vein surgery centers are based mostly on individual risk assessment by the way of implementation and extrapolation of general surgery VTE prophylaxis guidelines [4,11]. Simultaneously, the number and the quality of the studies dedicated to antithrombotic prophylaxis in open varicose vein surgery and minimal invasive treatment are very limited. In the presented study, the efficacy of routine VTE pharmacological thromboprophylaxis in patients undergoing saphenous vein stripping and miniphlebectomy was prospectively evaluated. Additionally, in the study population, the VTE risk factor evaluation was performed, and the utility of the Caprini score as a tool in the identification of the patients at highest risk of the postoperative DVT was investigated.

## 2. Material and Methods

This study was a prospective, observational study, projected on a cohort of surgical varicose vein patients. In the study, 141 patients undergoing saphenous vein stripping and miniphlebectomy while under spinal anesthesia were included. All the patients were treated in the same way, and in all of the cases, the same predefine study protocol was applied. 

The cohort of the study patients was selected from a group of 187 consecutive varicose vein patients admitted to the varicose vein surgical treatment. Patients with previous deep or superficial vein system thrombotic episodes were excluded from the study. Other exclusion criteria were the following: the use of anticoagulation, presence of leg ischemia or other contraindications to compression therapy, previous lower leg venous surgery and presence of the high risk of bleeding or contraindications to pharmacological thromboprophylaxis. According to the CEAP classification of the chronic venous disease based on the clinical (C), etiological (E), anatomical (A), and pathophysiological (P) factors, the study included 59 C2 patients, 48 C3, 18 C4, 10 C5 and 6 C6 patients, respectively. In all cases, venous duplex ultrasound (US) was performed, and varicose veins related to the great saphenous vein incompetence were diagnosed. The ultrasound (US) examinations of the superficial and deep lower leg vein system performed by the same physician were undertaken before the surgery and on the 10th and 30th day after surgery (together with the first preoperative evaluation and two scheduled postoperative controls). The ultrasound evaluation was blinded to the patients’ characteristics as well as knowledge regarding the risk factors. In the ultrasound examination, the superficial and deep vein systems of both lower legs were investigated, also including the ilio-caval segment. The presence of reflux, as well as thrombosis, including deep vein thrombosis, was assessed. Beyond CEAP classification, in the vein system disease severity evaluation, the Venous Clinical Severity Score (VCSS) was used. 

In all patients, concomitant disease presence and the presence of VTE risk factors were preoperatively evaluated. In the VTE risk assessment, the Caprini score was used (in all the patients, in the VTE risk evaluation, the presence of varicose veins, the scheduled surgery and its time were also included). Additionally, on the day before the procedure, during the preoperative patient preparation, preoperative D-dimer level evaluation was performed.

Intraoperatively, in all patients, after saphenous junction flash ligation, an invaginative stripping followed by miniphlebectomy was performed. All procedures were performed by the same surgeon. 

Postoperatively, elastic bandage compression was used for 1 day, followed by class 2 compression stocking for 6 weeks. In all patients, in their VTE prophylaxis, 40 mg enoxaparin once daily for 10 days was used (first dose 12 h before spinal anesthesia and surgery). Postoperative complications and complications related to the pharmacological thromboprophylaxis were assessed and reported. In the postoperative controls on day 10th and 30th, clinical signs potentially related to the clinically symptomatic deep vein thrombosis or symptomatic pulmonary embolism and lower leg circumferences were evaluated. 

The study endpoint was the presence of symptomatic or asymptomatic DVT confirmed in the imaging study (venous Doppler US). The study safety endpoint was major bleeding occurrence intraoperatively or within 30 days after surgery. 

The study was approved by the local Ethical Committee in Krakow (decision number 124/KBL/OIL/2017), and in all the patients, patient informed consent was obtained.

In the statistical analysis, the STATISTICA 13 PL (Statsoft Polska, Cracow, Poland) program was used. In the first stage, the parameters of descriptive statistics were determined. Student’s t-test was used to compare quantitative variables in the group with and without thrombosis, and the Mann–Whitney test in the case of non-normality. Normality was assessed with the Shapiro–Wilk test. Qualitative variables were assessed with the chi-square test. In further analysis, a logistic regression model was used to determine whether the analyzed variable is a risk factor for thrombosis. Based on the analyzed features, one-factor models were built for individual variables and a multi-factor model for significant variables. The odds ratio was calculated for each significant effect along with a 95% confidence interval. The correlation between the analyzed variables was assessed using the Spearman’s rank coefficient. In the all analyzes, the effects for which the probability value *p* was below the adopted significance level (α = 0.05) were assumed as significant.

## 3. Results 

In the study group, there were 107 female and 34 male patients with a median age of 49 years (from 18 to 84 years), and none of the patients was lost from follow-up or terminated the study before the end of the projected 30 day observation period. The preoperative patient group characteristics are presented in Table 1.

The mean duration of the surgical procedure was 48 min (from 25–80 min), and in the group, no surgical complications were noticed. Among the 141 patients who were operated on, the presence of a postoperative DVT was diagnosed in 5 cases (3.5%); all of these patients received a full 10-day course of pharmacological thromboprophylaxis with enoxaparin 40 mg od. In all of these cases, distal DVT in the same operated leg was confirmed with crural deep vein involvement. In one case, the presence of DVT was accompanied with clinical symptoms (symptomatic DVT in the study population: 0.7%). In the group with DVT cases, three out of five were diagnosed on day 10 postoperative control, while a further two were confirmed in the venous US examination performed 30 days after the procedure. No clinically documented PE episodes were noticed, and no bleeding complications nor any other adverse events potentially related to the pharmacological prophylaxis were observed. According to the Caprini model assessment, 4 patients (2.8%) were qualified as low VTE risk (young patients, short procedure, no additional risk factors), 85 presented moderate risk (60.2%), and in 52 cases (36.8%), high VTE risk was confirmed (the Caprini score distribution in the study population is presented in Figure 1). 

When comparing the patient groups with and without DVT, no cases of thrombophilia or previous cancer were noticed in the patients presenting with DVT. In the patients from the DVT group, none was taking oral contraceptives, and none was older than 74 years. In both groups (with and without DVT), a similar rate of obesity was reported, and in the positive cases in the DVT group, four out of five patients aged from 41 to 60 years were considered obese (Table 2). Regarding rheumatoid arthritis (three cases), psoriasis (two cases), and heart failure (five cases—stage I and II according to the New York Heart Association functional classification—NYHA), none of the five patients from the DVT-positive group had any of these conditions. There were also no statistically significant differences noticed concerning duration of the surgical procedure between the groups with and without DVT (Table 2).

In searching for other differences between the above-mentioned groups, significant variations in the Caprini score were noticeable (Table 2). When comparing patients with and without DVT, the median of the Caprini score in the subjects with diagnosed DVT episode was 5 and was significantly higher than in patients without DVT (median Caprine score: 4; *p* = 0.048). Among the five patients who developed DVT postoperatively, all had 5 points in their Caprini scores. Among the factors with statistical significance related to DVT occurrence, a higher CEAP advancement was also indicated. In patients with diagnosed DVT, the mean C was 4.6, whereas in the non-DVT, the mean C value was 2.92, with a statistically significant difference (*p* = 0.001). When patient groups with and without DVT were compared for VCSS score, higher scores were found for patients with DVT, and the median for patients with and without DVT was 10 and 8, respectively. The difference was statistically significant (*p* = 0.007).

Regarding the preoperative D-dimer level evaluation, a negative test result was confirmed in 119 patients, whereas in 22 cases, elevated D-dimer values were found. In the group of DVT patients, in 40% (two out of five patients), elevated preoperative D-dimer levels were reported. However, DVT was diagnosed in only 9.5% of all patients with initially elevated D-dimer levels. There was no statistically significant differences concerning preoperative D-dimer levels between the patients with and without DVT diagnosed in the postoperative course (median of the D-dimer level in the patients with DVT: 462 (239–3200) µg/L vs. median of the D-dimer level in the patients without DVT: 265 (8–1914) µg/L; *p* > 0.05). The distribution of the D-dimer level values is presented in Figure 2.

In a further step of the analysis, logistic regression analysis was used to assess the statistical significance of the identified potential VTE risk factors in the study population. In the univariable assessment, the following variables were included: age; sex; BMI; surgery duration; Caprini score; VCSS; CEAP class; and laboratory data, including preoperative D-dimer level, hematocrit, hemoglobin level and platelet count. There were no cases of thrombophilia, cancer history or chronic heart failure among the patients with diagnosed DVT. None of these patients was also on oral contraceptive treatment. Age, patient gender and obesity were not observed to be statistically significant. As previously confirmed in the group comparison and in the logistic regression model, the result of the Caprini score evaluation was observed to be a statistically significant predictor for DVT occurrence (*p* = 0.0485), with odds ratio (OR) = 2.04 (95% CI = (1.005–4.105)). Another factor that became statistically significant in terms of postoperative DVT prediction was the reported Venous Clinical Severity Score rate (*p* = 0.007). The calculated odds ratio for VCSS results was 1.98 (95% CI (1.19; 3.26)). Contrary to this observation, the preoperative D-dimer level had a very low influence on the DVT occurrence, with OR = 1.001 (95% CI (1.00–1.003)). Moreover, when adjusting the estimated D-dimer levels to the age of the patients (adding 100 ug/L to the cut of value for every 10 years over 50), there was no significant influence of this parameter on the DVT prevalence.

Multiple logistic regression analysis was then applied to assess the combined effect of the evaluated parameters on DVT occurrence. Combining the previously mentioned VTE risk factors and other evaluated parameters in the multifactor model, the patient age (OR = 0.86; 95% CI (0.75–0.99)), Caprini score evaluation results (OR = 4.04; 95% CI (1.26–12.9)) and VCSS results (OR = 2.4.; 95% CI (1.23–4.7)) were statistically significant as predictors for postoperative DVT, with a *p* value of 0.029 for age as well as *p* = 0.017 and *p* = 0.009 for Caprini score results and VCSS results, respectively. In the analysis, we could not confirm the role played by the other factors, including the preoperative D-dimer value in the combined model in the study cohort; however, the small number of the DVT events suggests very careful interpretation of the results. The potential influence of the patient group size and especially the low DVT prevalence did not allow us to validate the efficacy of the Caprini model in our study cohort, but the presence of the Caprini score results among the identified factors related to the postoperative DVT occurrence in both univariable analysis as well as multiple regression analysis suggest the potential link among the factors included in this score and DVT development. 

## 4. Discussion

In the reported study, the efficacy of routine thromboprophylaxis in a group of patients who underwent saphenous vein stripping and phlebectomy was assessed. The presence of a postoperative DVT was diagnosed in 3.5% of cases, despite a 10-day course of low molecular weight heparin (LMWH) pharmacological prophylaxis. Among the statistically significant variables corresponding to DVT occurrence, the results of the Caprini score evaluation were identified. Among other factors related to the higher coincidence with postoperative DVT, class C of the CEAP classification and the VCSS results were identified. Both of them are not included in the Caprini score form.

DVT remains one of the most significant complications of varicose vein surgical treatment. Despite the high number of patients that still undergo this kind of the invasive treatment in many countries, the prospective studies dedicated to the pharmacological prophylaxis in this clinical scenario are very limited in number. In the prospective randomized controlled study on a cohort of 262 patients with moderate DVT risk operated on for varicose veins, San Norberto Garcia et al. did not confirm ultrasonographically any DVT cases in a subgroup receiving a 10-day LMWH course or in a subgroup without pharmacological perioperative thromboprophylaxis [19]. The authors of this study conclude that in patients with moderate VTE risk who had undergone varicose vein surgery with early mobilization and postoperative compression, there is no need to use routine pharmacological thromboprophylaxis.

In a study dedicated to varicose vein surgery patients, Wang and coworkers used four different approaches to perioperative thromboprophylaxis, including a subgroup not receiving prophylaxis (the postoperative DVT rate and PE rate in this subgroup being 5.17% and 1.48%, respectively), as well as a subgroup receiving LMWH in doses of 4000 IU twice a day (with a 0.36% DVT rate and a 0% PE rate) [20]. In the remaining two groups, unfractionated heparin in three prophylactic doses a day was used (with a DVT rate of 0.56% and a PE rate of 0%), or LMWH in a dose of 6000 U once a day was applied (with DVT and PE rates of 0.35% and 0%, respectively). The authors’ conclusion suggest the potential role of pharmacological thromboprophylaxis in effectively reducing DVT and PE rates in patients undergoing saphenous vein stripping [20].

In another prospective study on VTE prophylaxis in patients undergoing varicose vein surgery, the influence of the duration of pharmacological prophylaxis was investigated. In this study, LMWH courses of 2 and 7 days were compared. In the resulting analysis, no significant differences between short and longer durations of LMWH prophylaxis were noticed [21]. 

Due to methodological and group characteristic variations, the lack of high quality data regarding proper VTE prophylaxis in this population still does not allow for the proposal of a definitive thromboprophylaxis algorithm regarding both drug administration and prophylaxis duration. Early or delayed mobilization, type of anesthesia used, the use of compression, individual risk factor presence, and a variety of treated pathologies and venous system abnormalities suggest rather an individual approach to VTE risk assessment and its prophylaxis implementation in this particular patient population [4]. Extrapolation of the surgical guidelines, as well as if available, the particular venous center or surgical department thromboprophylaxis protocol, should be always based on an individual approach to both patient- and procedure-related VTE risk factors. In our study, a hospital protocol dedicated to varicose vein surgery (great saphenous vein stripping and miniphlebectomy) was set out. Herein, the surgery was performed using spinal anesthesia, with pharmacological thromboprophylaxis (enoxaparin, 40 mg) being started before the procedure (12 h before the anesthesia), and pharmacological prophylaxis being continued for up to 10 postoperative days. Despite this prophylactic approach, in the cohort of 141 patients, 5 DVT cases were noticed, including one case of symptomatic DVT. As documented in other studies dedicated to general surgery patients, none of the available VTE prophylaxis methods can completely eliminate the risk of DVT [22,23,24,25,26]. However, most of the performed studies confirm that the use of pharmacological thromboprophylaxis can significantly reduce the postoperative VTE rate or its clinical significance [11,12,22,23,24,25,26]. In our material, in all cases, only distal DVT (mostly asymptomatic) was indicated—no cases of clinically documented PE were reported. In regard to the duration of the pharmacological thromboprophylaxis, it should be emphasized that two out of five postoperative DVT cases were diagnosed in 30 days, but not in the 10-day postoperative control (after stopping LMWH prophylactic administration).

According to the individual risk assessment, among the patients in our study group, 60.2% were at moderate risk and 36.8% were at high risk of VTE development. When applying the Caprini score criteria, all five patients with US-confirmed DVT obtained 5 points in the Caprini score, which means that all of them were in the high-risk group. Among the remaining 132 patients from the high-risk group and moderate-risk group, the applied thromboprophylaxis was observed to be clinically successful (including the patients with a reported Caprini score of up to 8 points). 

In searching for other factors potentially related to DVT development, despite the applied thromboprophylaxis, we focused on the specific disease (chronic venous disease, CVD)-related factors. In the subgroup of DVT patients, after varicose vein surgery, a higher mean C class (according to the CEAP classification) was noticed. Another factor that became statistically significant as a postoperative DVT predictor was the Venous Clinical Severity Score rate, with an OR for VCSS results of 1.98. The coincidence of the higher C class and higher VCSS with DVT occurrence suggests a possible link between chronic disease advancement and its severity and higher DVT risk. The clinical significance of this relationship should, however, be confirmed in large epidemiological studies. According to previous publications, the presence of varicose veins is one of the most important risk factors for deep vein thrombosis occurrence. Chang et al., in a study based on 211,984 varicose vein patients and a similar size group without varicose veins (with median follow-up at 7.5 years), noted that the varicose vein group had higher incidence rates for DVT than the control group (6.55 vs. 1.23 per 1000 person/year) [27]. The presence of varicose veins as a DVT risk factor was also documented in hospitalized patients (both medical as well as surgical) [28]. Undoubtedly, varicosities are also perceived as a risk factor for lower leg superficial vein thrombosis (SVT) [29]. Of note, SVT, according to the current knowledge, is no longer considered to be a benign disease only. As documented in the literature, a significant number of SVT patients have also been co-diagnosed with symptomatic or (more often) asymptomatic DVT or PE [30,31]. In our study, no SVT cases in the preoperative US were noticed, and the presence of a previous venous thrombotic episode (superficial or deep) remained among the exclusion criteria.

When comparing the groups of patients with and without DVT, the statistically significant importance of the individual Caprini score results was noticed. In the patients with DVT episode despite pharmacological prophylaxis, a significantly higher Caprini score was found. The importance of Caprini score result assessment was also noticed in the logistic regression model analysis as a statistically significant predictor for DVT occurrence in our study group (with OR = 2.04). Moreover, in a multiple factor model analysis, the Caprini score results, together with the patient age and VCSS assessment results, show statistically significant importance as postoperative DVT occurrence predictors.

The results concerning the Caprini score should be interpreted with caution. According to our study results, the efficacy of the Caprini score evaluation was observed to be limited in high-risk patient identification in the study population. Moreover, as also documented in our study, some other potential factors could also be part of clinical VTE risk evaluation in these particular settings. As previously stated, to date, there has been no validation study of the Caprini score in a population of varicose veins patients undergoing saphenous vein open surgery. The heterogeneity of this patient group in regard to patient characteristics, anesthesia, surgical procedure and perioperative care will probably make the performance of such a study very challenging. In addition, in our study, the number of patients in the study cohort together with the low number of DVT events does not allow one to perform correct validation of the Caprini score or to propose the new valid prognostic model. Currently, at Pirogov University in Moscow, Russia, the prospective validation of the Caprini score in surgical varicose vein patients is being performed (ClinicalTrials.gov: NCT03041805). To reach the goals of proper statistical analysis evaluation, in this ongoing trial, a study population of 3000 varicose vein surgical patients was projected.

Despite the limited knowledge and research results, the Caprini score is used in varicose vein surgical patients in several phlebology centers. It should also be emphasized that in many countries, surgery remains one of the basic treatment methods for varicose vein patients, justifying the study’s focus on proper VTE risk assessment in this population. An important implication corresponding to the results ofour observation, as well as to the other studies dedicated to this described clinical situation, is the confirmation of the necessity of an individual VTE risk assessment in these patient cohorts [4]. Additionally, in the VTE risk evaluation in this particular patient population, other potential factors, including the CEAP C class, as well as CVD severity (VCSS evaluation), should probably be taken into consideration (these are not included into the current version of the Caprini score) [13].

Further studies are needed to propose the score modifications or another model of the objective and validated VTE risk assessment in this clinical situation. 

The preoperative D-dimer evaluation is usually not routinely performed in varicose vein patients. Moreover, when performed, its diagnostic and prediction values were often questioned [32,33,34]. Similarly to these observations, we could not confirm the potential prediction value of the preoperative D-dimer levels in our study cohort, even when the elevated D-dimers were preoperatively noticed in a significant number of patients. We know that D-dimers are not specific and can be elevated in patients with various conditions including inflammation (which occurs more often in more advanced chronic venous disease stages), as well as in patients with concomitant diseases and in the older populations. In previous studies dedicated to coagulation system disturbances, prothrombotic state marker presence in more advanced stages of chronic venous disease was also reported [35,36,37,38].

The diagnosed distal DVT was mostly asymptomatic (in four out of five patients with confirmed DVT). However, it should also be emphasized that distal DVT cannot be free from early and late complications, including pulmonary embolism or post-thrombotic syndrome. In the Contention Alone Versus Anticoagulation for Symptomatic Calf Vein Thrombosis Diagnosed by Ultrasonography (CACTUS PTS) study, the postthombotic syndrome prevalence in patients with isolated calf DVT after median follow-up of 6 years was 30% [39]. 

Some study limitations should also be mentioned in regard to our research. The study was observational only and no group comparison or randomization was applied (to compare the patients with and without prophylaxis). In addition, the study was performed according to the hospital thromboprophylaxis protocol dedicated to this particular patient group, and there was no hospital committee agreement to perform the spinal anesthesia and varicose vein open surgery without thromboprophylaxis. Another important limitation of the study is related to the size of the study cohort, which suggests very careful result interpretation, especially in the light of the low rate of postoperative DVT prevalence in the patients undergoing VTE thromboprophylaxis. The authors are aware that the Caprini score validation, as well as new prediction model proposals, should be based on a bigger population and on a bigger number of reported DVT events. The descriptive and explorative nature of the achieved results in our study does not disqualify the Caprini score in this particular clinical settings, but suggest the need of the continuation of the further research on the valid and effective model of VTE risk assessment in these particular clinical settings. In all of the patients in our cohort, an active VTE prophylaxis was used according to the hospital thromboprophylaxis policy, and the reported incidence of DVT corresponds to the results of the few other available prospective observations in varicose vein surgery patients. Further studies are needed to identify the patients who benefit most from antithrombotic prophylaxis in varicose vein surgery. In the study, we also did not check the postoperative D-dimer values on the 10th and 30th day after procedure, which, however, together with postoperative hematoma presence, as well as inflammation related to the healing process, would be difficult to objectively interpret.

## 5. Conclusions

All the patients undergoing varicose vein surgery should undergo VTE risk evaluation based on the individual assessment.In VTE risk evaluation, patient and surgical procedure characteristics based on the factors included into the Caprini score but also on specific chronic venous disease related factors should be taken into consideration.Further studies are needed to propose an objective and validated VTE risk assessment model, as well as a validated antithrombotic prophylaxis protocol in this particular patient group.

## Figures and Tables

**Figure 1 jcm-09-03970-f001:**
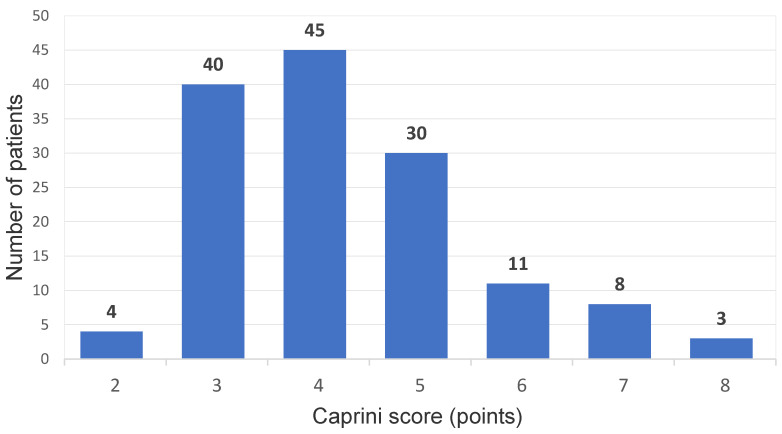
Caprini score results in the study population.

**Figure 2 jcm-09-03970-f002:**
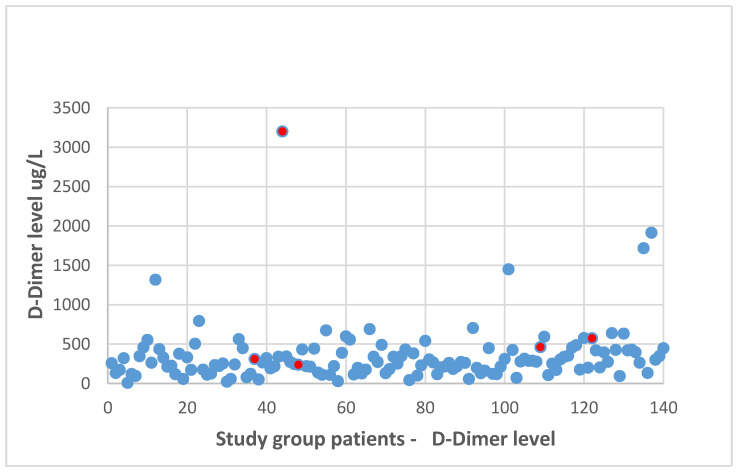
Preoperative D-dimer levels in the whole study group (the levels in five patients with confirmed DVT marked).

**Table 1 jcm-09-03970-t001:** Characteristics of the study population—preoperative evaluation of the concomitant disease as well as venous thrombo-embolism risk factor presence.

Concomitant Diseases and VTE Risk Factors	No of Patients/%
Overweight (BMI> 25 kg/m^2^)	88 (62.4%)
Obesity (BMI > 30 kg/m^2^)	69 (48.9%)
Arterial hypertension	42 (29.7%)
Oral contraceptive use	9 (6.38%)
Diabetes	9 (6.38%)
Cancer (non-active/after treatment)	5 (3.5%)
Chronic heart failure (NYHA I or II)	5 (3.5%)
Heart ischemic disease	4 (2.8%)
Hyperthyreosis	4 (2.8%)
Asthma	4 (2.8%)
Rheumatoid arthritis	3 (2.1%)
Psoriasis	2 (1.4%)
Thrombophilia	1 (0.7%)
Age characteristics (y)	
≤40	37(27.8%)
41–60	69(47.5%)
61–74	29 (20.5%)
≥75	6 (4.2%)

BMI–Body Mass Index; NYHA–New York Heart Association functional classification, VTE–venous thromboembolism.

**Table 2 jcm-09-03970-t002:** Group comparison in regard to the potential identified VTE risk factor presence: 136 patients without postoperative deep vein thrombosis (DVT) vs. DVT positive group.

Risk Factor	Patients without Postoperative DVT	Patients with Postoperative DVT	Level of Statistical Significance of the Group Comparison
Age: 41–60 y	47.7%	80%	NS
Age: 61–74 y	20.5%	20%	NS
Obesity	48.5%	60%	NS
Female gender	76.4%	60%	NS
Surgery duration	48.6 ± 11.1 min	51 ± 5.7 min	NS
Diabetes	6.6%	0%	NS
Caprini score result (median)	4 (2–8)	5 (al patients 5 points)	*p* = 0.048
C class according to CEAP classification	2.92 ± 1	4.6 ± 0.8	*p* = 0.001
VCSS result (median)	8 (6–15)	10 (9–11)	*p* = 0.007

NS—nonsignificant.

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
