# Peer review of "Venous Thromboembolism Prophylaxis and Thrombotic Risk Stratification in the Varicose Veins Surgery—Prospective Observational Study"

_jcm, 2020, doi:10.3390/jcm9123970_

Round 1

Reviewer 1 Report

the requested changes have been made to the text. I have no additional questions or commentaries. 

Author Response

The Reviewer has already accepted previous revisions.

Reviewer 2 Report

The authors have made adjustments to the manuscript according to my previous comments but unfortunately the conclusion in the abstract has not been adjusted to align with the conclusion in the manuscript. The main study limitations (low effective sample size) are not resolved (this would of course be impossible without collecting more data).

Author Response

Dear Reviewer,

Once again thank You for all the advices, which allowed us to improve the text – we are aware about the study limitations,  but it is always difficult to find the resources and possibilities to work on very big number of patients from the very particular and homogenous group, which would be of course great to collect. It is also not always possible in every day clinical practice while working in the hospital (the study projected on validation of the Caprini score in Russia plans to recruit 3000 patients within several years – unfortunately, not all of us have such possibilities to conduct such an extensive study). 

As suggested in Your last review we changed the conclusion in the abstract to correct it in accordance with the conclusion in the manuscript. We also introduced into the abstract the information about the lack of the possibility of the Caprini score validation (due to the study population size and number of  DVT events).  

Once again thank for all important comments.

Kind regards,

On behalf of the authors, Maciej Wołkowski

Reviewer 3 Report

Authors assessed symptomatic and asymptomatic DVT who underwent varicose vein surgery, and described that VTE risk assessments, Caprini score result and VCSS results, are able to evaluate the risk of DVT even after varicose vein surgery. Manuscript is well written, and authors described all of my concerns. 

Author Response

The Reviewer has already accepted previous revisions.

This manuscript is a resubmission of an earlier submission. The following is a list of the peer review reports and author responses from that submission.

Round 1

Reviewer 1 Report

Caprini's score seems to be far too exhaustive and is difficult to perform in practice. In particular, it includes blood tests for thrombophilia, which are obviously not feasible in all patients.

In the material and methods chapter, the diagnosis of deep vein thrombosis on ultrasound is poorly specified. It is mentioned in the article « looking for a reflux or potentially thrombotic modification ». This is not a validated endpoint of venous thrombosis as compression. Has an exploration of the iliocave segment been carried out systematically ?

We do not understand the relevance of peri operative D-dimers. It is not specified whether patients with high levels of preoperative D-dimers have suffered vein thrombosis more than others.

The average duration of the surgical procedure was 48 minutes, in addition to the duration of the setting up of spinal anesthesia. Doesn't this duration of intervention justify the systematic administration of thrombosis prophylaxis?

In this study most patients had Caprini corresponding to a moderate (60%) or high risk (36.8%) of thrombosis and very few havec low risk (2.8%). Indeed what is the additional vlaue of the Caprini score if according of the risk, administration of prophylaxis thrombosis should be systematic and not questionnable.

It is not specified whether thrombosis was homolateral or controlaral to the operated side.

It is not clear whether deep vein thrombosis occurs in longer interventions. These interventions could last 25 to 60 minutes.

It is not specified whether deep vein thrombosis occurs in stripping and phlebectomy.

There is no mention of the adverse effects of thromboprophylaxis in terms of bleeding.

Author Response

Dear Reviewer,

Thank you for your comments and suggestions. The answers for your questions are written below.

1.Caprini's score seems to be far too exhaustive and is difficult to perform in practice. In particular, it includes blood tests for thrombophilia, which are obviously not feasible in all patients.

Answer:

Caprini score evaluation is based on the physical examination as well as information obtained from anamnesis, patient history as well as information related to the current treatment and procedures such as surgery. No need for routine laboratory thrombophilia screening when using Caprini score but the points are given for diagnosed thrombophilia (if such information is available in the patients history).

In several centers in our country as well as other regions, Caprini score is used in the VTE risk assessment. Of course it is not obligatory and used only in some places. On the other hand there are very few validated alternatives available so many people choose Caprini model. In other centers, as we can observe in the clinical practice, VTE risk evaluation is based on the experience and knowledge of the physician. Caprini score was validated only in some specialities such as surgery or intensive care. Unfortunately, no validation in varicose vein surgery or vascular surgery are available. Our research brings new information about the clinical significance of the VTE risk factors which have been included into this score. The study is not Caprini score validation in varicose vein surgery patients but gives new data concerning the significance of the Caprini score including VTE risk factors in this clinical situation. Moreover, the study identified the risk factors not included into the score.

2.In the material and methods chapter, the diagnosis of deep vein thrombosis on ultrasound is poorly specified. It is mentioned in the article « looking for a reflux or potentially thrombotic modification ». This is not a validated endpoint of venous thrombosis as compression. Has an exploration of the iliocave segment been carried out systematically ?

Answer:

In the cases ilio-caval segment was also evaluated. The sequence regarding ultrasound examination and possible thrombotic changes ( deep and superficial vein thrombosis presence – acute, as well as previous thrombotic episodes) was modified in the manuscript text (marked in the text) to emphasize that both reflux presence as well as thrombosis presence or absence were investigated.

3.We do not understand the relevance of peri operative D-dimers. It is not specified whether patients with high levels of preoperative D-dimers have suffered vein thrombosis more than others.

Answer:

There was no statistically significant difference concerning DVT occurrence between the patients with elevated preoperative D-Dimer level and the patients with D Dimers below the cut off value – this information was added to the manuscript text. Simultaneously, in the patients with elevated D Dimer level we did not observe the higher rate of the DVT cases. There was a mistake in line 187 – the correct results are: 40% patients with DVT had elevated D-dimers (2 of 5, not 3 of 5) .

4.The average duration of the surgical procedure was 48 minutes, in addition to the duration of the setting up of spinal anesthesia. Doesn't this duration of intervention justify the systematic administration of thrombosis prophylaxis?

Answer:

In all of the patients the same protocol of prophylaxis was applied (Independent of varicose vein surgery time duration). In our hospital we use VTE pharmacological thromboprophylaxis in all varicose vein patients operated under spinal anesthesia – this protocol was approved by the hospital VTE prophylaxis committee (there are no special guidelines available in the literature which are based on EBM and dedicated to this particular group of the patients). According to the hospital thromboprophylaxis protocol, we also do not use routinely Caprini score in the patient evaluation as there is no previous validation of this score in varicose vein surgery available). The use of the Caprini score was the part of the current  study protocol to assess the efficacy of this score in the identification of VTE factors (as well as to find other factors which are present but not included into the Caprini score).  

5.In this study most patients had Caprini corresponding to a moderate (60%) or high risk (36.8%) of thrombosis and very few havec low risk (2.8%). Indeed what is the additional vlaue of the Caprini score if according of the risk, administration of prophylaxis thrombosis should be systematic and not questionnable.

Answer:

You are fully right and we completely agree with this. This suggestion is in accordance with our study results and study conclusion nr 3. Caprini score is probably not the best tool to identify the patients with the highest risk of DVT occurrence in the group of patients undergoing varicose vein surgery. As we mentioned in the text there was no validation of this score in varicose vein surgery. The results of our research suggest that in this particular group, in the VTE risk assessment, some other potential VTE risk factors should be included. Currently we still use routine prophylaxis in our varicose vein patients but as we suggest in the discussion and conclusion the new research towards the more effective and validated model of VTE risk assessment in this population should be performed. We added the information about the inefficacy of the Caprini score as the only method of VTE risk assessment in the patinets with varicose veins to the manuscript text.

6.It is not specified whether thrombosis was homolateral or controlaral to the operated side.

Answer:

DVT always affected the same limb that had been operated.DVT was always ipsilateral and was diagnosed in the same lower leg – this information was added to the text

7.It is not clear whether deep vein thrombosis occurs in longer interventions. These interventions could last 25 to 60 minutes.

Answer:

There was no statistically significant difference regarding the time of the surgery between the patients with and without DVT – this information was added to the text in the separate sentence.

8.It is not specified whether deep vein thrombosis occurs in stripping and phlebectomy.

Answer:

In all the patients the same procedure was performed: stripping and phlebectomy.

9.There is no mention of the adverse effects of thromboprophylaxis in terms of bleeding.

Answer:

We did not observe any adverse effects of thromboprophylaxis – this information is included into the text  .

Reviewer 2 Report

The authors have conducted a prospective study assessing the efficacy of routine VTE prophylaxis, and the incidence of VTE after varicose vein striping. In addition they aimed to assess the performance of the Caprini VTE risk score. Although I have read the article with interest, and I like the general design and see the relevance of the question, there are several major concerns with the manuscript and a re-analysis of the data seems necessary.

Major comments:

  • Introduction: the overall aims would benefit from a more clear description; I do not see what the evaluation of efficacy of prophylaxis has to do with the Caprini risk score. Suggest rephrasing
  • Methods: please describe whether the study was conducted following according to an a priori defined protocol; explicate the study design; describe whether and when patients were consecutively included (or why not); describe how the outcome assessment took place and whether this was blinded to patient characteristics; describe the dependent variables that were considered for inclusion in the statistical model; describe all statistical methods that were used (ie t-test etc); describe the cut-off or decision making for including variables in the multivariate models; and describe considerations on missing data and lost-to-follow up.
  • Methods: the age categories in the table with baseline characteristics do not add up to 141
  • Methods: this section contains text that belongs to the results section (ie description of the sample). I suggest moving table 1 and all descriptions of the population to the results section.
  • Methods / results: the methods applied are not sufficient for evaluation of the performance of the Caprini score. At the very least you should provide information on calibration and discrimination (C statistic).
  • Results: Table 2 is not clear to me; please consider to stratify data according to each group
  • Results: The comparison between median Caprini scores with P = 0.0485 has a CI of 0.998 – 4.18 which is impossible. Either the P-value should be > 0.05 or the CI lower boundary > 1. Either way, it suggests the difference between groups may very well be only marginal.
  • Results: describe whether there was lost-to-follow-up and missing data.
  • Results: provide odds ratio’s with 95% CI for all logistic regression results, not only P -values
  • Results: Figure 2 is a very odd way of showing D-dimer levels. In contrary to your phrasing it does not show the distribution because, if I understand correctly, the X-axis are the individual patients? A histogram with D-dimer categories or just a mean with SD (or median+IQR) should suffice to provide an impression of the distribution.
  • Results: I do not see the relevance of the correlations described in figure 3 and 4?
  • Discussion: consider starting with a brief summary of your main findings
  • Discussion: the first conclusion is too strongly worded and cannot be derived from the data at hand without additional analyses.
  • Please elaborate in the discussion on the clinical relevance of distal vein thrombosis (as opposed to proximal vein thrombosis)
  • Please add discussion on methodological limitations (for example the limited sample size with low event rate)

Minor comments:

  • The introduction is quite comprehensive which is nice but it would benefit from more focus (ie shortening) on the main issue which is the incidence of and risk factors for VTE after varicose vein surgery.
  • 106 Suggest rephrasing ‘experimental section’ to ‘Methods’
  • 114 Please provide a short explanation of the “CEAP” classification which may be unknown to the average reader.

Author Response

Dear Reviewer,

Thank you for your comments and suggestions. The answers for your questions are written below.

The authors have conducted a prospective study assessing the efficacy of routine VTE prophylaxis, and the incidence of VTE after varicose vein striping. In addition they aimed to assess the performance of the Caprini VTE risk score. Although I have read the article with interest, and I like the general design and see the relevance of the question, there are several major concerns with the manuscript and a re-analysis of the data seems necessary.

Major comments:

1.Introduction: the overall aims would benefit from a more clear description; I do not see what the evaluation of efficacy of prophylaxis has to do with the Caprini risk score. Suggest rephrasing.

Answer:

Thank You for this comment – the rephrasing was performed emphasizing two main goals of the study: the evaluation of the routine thromboprophylaxis in the patients undergoing varicose vein surgery under the spinal anesthesia as well as the utility of the Caprini score as a tool which identified the patients at highest risk of postoperative DVT. There is no validated model of VTE risk yet as well as there is no validated VTE prophylaxis protocol dedicated to the patients undergoing varicose vein surgery  available. In some phlebology and surgery centers, Caprini score is used (extrapolating the experience from general surgery studies), but in reality, no validated and efficient model of VTE risk assessment in this particular population is known. There is no validation of the Caprini score in the varicose vein surgery in any of the previous studies available. An introduction of the Caprini score evaluation as the part of the study should answer the question - If this particular VTE risk evaluation model is effective in the identification of the patients with highest risk of DVT development or  perhaps another model of the risk assessment should be proposed?.   

2.Methods: please describe whether the study was conducted following according to an a priori defined protocol; explicate the study design; describe whether and when patients were consecutively included (or why not); describe how the outcome assessment took place and whether this was blinded to patient characteristics.

Answer:

The study was prospective and was performed according to the predefined protocol – we modified the section describing “methods” in the manuscript text to specify the steps of the protocol in the more specific way. Regarding the patient population we included consecutive patients except the individuals where the exclusion criteria were found (the exclusion criteria were specified in the manuscript text) .

All the patients were operated by the same team of the surgeons and in all the cases the ultrasound evaluation was performed by the same physician (in the same way as well as using the same ultrasound device). The ultrasound evaluation was blinded to the patients characteristic as well as knowledge regarding the risk factors  - this information was included into the manuscript text)

3.Describe the dependent variables that were considered for inclusion in the statistical model; describe all statistical methods that were used (ie t-test etc).

Answer:

The part of the text regarding the statistics was improved and marked in the text

“During the statistical analysis, the STATISTICA 13 PL program was used. In the first stage, the parameters of descriptive statistics (mean, standard deviation, median, and minimum and maximum values) were determined. Student's t-test was used to compare quantitative variables in the group with and without thrombosis, and the Mann-Whitney test in the case of non-normality. Normality was assessed with the Shapiro-Wilk test. Qualitative variables were assessed with the Chi-square test. In further analysis, a logistic regression model was used to determine whether the analyzed variable is a risk factor for thrombosis. Based on the analyzed features, one-factor models were built for individual variables and a multi-factor model for significant variables. The odds ratio was calculated for each significant effect along with a 95% confidence interval. The correlation between the analyzed variables was assessed using the Spearman's rank coefficient. In the all analyzes, the effects for which the probability value p was below the adopted significance level α = 0.05 - were assumed as significant.”

4.Describe the cut-off or decision making for including variables in the multivariate models; and describe considerations on missing data and lost-to-follow up.

Answer:

There was no patient lost from follow up – so, in the analysis all the patients and their data have been included. In the multivariate analysis, the construction of the model was started with a larger number of parameters using the "step down” method and removing the irrelevant parameters one by one.The combination of the parameters that remained significant and did not destroy the sense of the model were left. 

5.Methods: the age categories in the table with baseline characteristics do not add up to 141

Answer:

We discounted the patients below 40 in the table I as there is no point for the age in this group in the Caprini score. According to Your suggestion we added the data ragarding the patients to the table I and study population characteristics.

6.Methods: this section contains text that belongs to the results section (ie description of the sample). I suggest moving table 1 and all descriptions of the population to the results section.

Answer:

We modified the text moving the suggested part to the result section.

7.Methods / results: the methods applied are not sufficient for evaluation of the performance of the Caprini score. At the very least you should provide information on calibration and discrimination (C statistic).

Answer:

The C static have been added. For the Caprini score C value of 0.81 was calculated.

The information have been added into the text.

8.Results: Table 2 is not clear to me; please consider to stratify data according to each group

Answer:

The table was modified and the data concerning each group were included

9.Results: The comparison between median Caprini scores with P = 0.0485 has a CI of 0.998 – 4.18 which is impossible. Either the P-value should be > 0.05 or the CI lower boundary > 1. Either way, it suggests the difference between groups may very well be only marginal.

Answer:

Thank You for this comment. We checked the statistics results and found the mistake regarding the values presented in the manuscript text. The correct value is  P=0,0485, OR = 2,042 and CI of 1.005 - 4,1505) .The previously mentioned CI was our mistake made during editing the text.

10.Results: describe whether there was lost-to-follow-up and missing data.

Answer:

No missing data in analysis, no lost from follow up – all the patients completed the follow up period.

11.Results: provide odds ratio’s with 95% CI for all logistic regression results, not only P –values.

Answer:

We added the missing data regarding OR for CEAP

CEAP: OR = 2,79  95%CI (1.36 – 5.74)

12.Results: Figure 2 is a very odd way of showing D-dimer levels. In contrary to your phrasing it does not show the distribution because, if I understand correctly, the X-axis are the individual patients? A histogram with D-dimer categories or just a mean with SD (or median+IQR) should suffice to provide an impression of the distribution.

Answer:

On x axis individual patient results are present (this information was introduced into the figure description). We added to the manuscript text information about the median of D -Dimer levels.

Median of D-dimer level in the patients without DVT 265 (8 -1914)

Median of D-dimer level in the patients with DVT 462 (239-3200)

13.Results: I do not see the relevance of the correlations described in figure 3 and 4?

Answer:

In both of the cases, weak (but statistically significant) correlation was observed. Preoperative D Dimer level did not influence directly on the DVT prevalence in the study population but correlated with the Caprini score. The same observation was suggested regarding Venous Clinical Severity Score and D dimer values. Both of these parameters (Caprini score as well as VCSS) have been identified as the risk factors of DVT occurrence in our study, which can suggest an indirect link between the D dimer level and thrombosis occurrence. Of course this observations should be interpreted with caution – we know that D dimers are not specific and can be elevated in the patients with inflammation (which happen more often in more advanced chronic venous disease stages with usually higher VCSS rate as well as in the patients with concomitant diseases and in the older populations) – this remarks have been introduced into the  discussion text.

.

14.Discussion: consider starting with a brief summary of your main findings.

Answer:

The discussion was modified and a brief summary of main finding was moved into the beginning of the discussion.

15.Discussion: the first conclusion is too strongly worded and cannot be derived from the data at hand without additional analyses.

Answer:

Thank You for this suggestion. We changed the conclusion 1 from:

“Individual risk assessment allows selecting out patients at risk of VTE development in patients undergoing surgical varicose vein treatment.”

to

“All the patients undergoing varicose vein surgery should undergo VTE risk evaluation based on the individual assessment.”

16.Please elaborate in the discussion on the clinical relevance of distal vein thrombosis (as opposed to proximal vein thrombosis)

Answer:

The information about the clinical relevance of the distal DVT have been added to the manuscript text.

17.Please add discussion on methodological limitations (for example the limited sample size with low event rate).

Answer:

The information about the study limitation  was added to the discussion

Minor comments:

1.The introduction is quite comprehensive which is nice but it would benefit from more focus (ie shortening) on the main issue which is the incidence of and risk factors for VTE after varicose vein surgery.

Answer:

We shortened the introduction to the most important data focusing on VTE risk assessment, varicose vein surgery and its related VTE risk.

2.106 Suggest rephrasing ‘experimental section’ to ‘Methods’.

Answer:

Thank You, we changed it into “methods”

3.114 Please provide a short explanation of the “CEAP” classification which may be unknown to the average reader.

Answer:

We supplied the text with an explanation of  the term of CEAP classification.

Reviewer 3 Report

Comments to Authors

Authors assessed symptomatic and asymptomatic DVT who underwent varicose vein surgery, and described that VTE risk assessments, Caprini score result and VCSS results, are able to evaluate the risk of DVT even after varicose vein surgery. They concluded that patient and surgical procedure characteristics based on the factors included into the Caprini score but also on specific chronic venous disease related factors should be taken into consideration. The manuscript is well written; however, I have some comments for this article to publish in Journal of Clinical Medicine.

Major Concerns:

  1. I have great some concerns about statistical analysis. At first, only 5 out of 141 operated patients were developed DVT. The number of DVT patients were very small and I think that logistic regression analysis cannot be performed with this number of events. Second, in Figure 3 and Figure 4, authors described that Caprini score and VCSS were correlated with preoperative D-dimer. In fact, p value is less than 0.05, however, r is very low, only 0.30 and 0.21. These values suggest that coefficient of determination (R2) is too low (0.09 and 0.0441) to make sense as a predictive model.
  2. I found that the study objectives are ambiguous in this study. Did you intended only to assess the frequency of DVT after varicose vein surgery? VTE risk score such as Caprini score is established to evaluate the incidence of DVT after surgery. Do varicose vein surgery have extra higher risk of DVT than other surgery? Clinical implication of this study is unclear.

Author Response

Dear Reviewer,

Thank you for your comments and suggestions. The answers for your questions are written below.

Authors assessed symptomatic and asymptomatic DVT who underwent varicose vein surgery, and described that VTE risk assessments, Caprini score result and VCSS results, are able to evaluate the risk of DVT even after varicose vein surgery. They concluded that patient and surgical procedure characteristics based on the factors included into the Caprini score but also on specific chronic venous disease related factors should be taken into consideration. The manuscript is well written; however, I have some comments for this article to publish in Journal of Clinical Medicine.

Major Concerns:

1.I have great some concerns about statistical analysis. At first, only 5 out of 141 operated patients were developed DVT. The number of DVT patients were very small and I think that logistic regression analysis cannot be performed with this number of events. Second, in Figure 3 and Figure 4, authors described that Caprini score and VCSS were correlated with preoperative D-dimer. In fact, p value is less than 0.05, however, r is very low, only 0.30 and 0.21. These values suggest that coefficient of determination (R2) is too low (0.09 and 0.0441) to make sense as a predictive model.

Answer:

Thank You for this review and remarks.

We do agree that the rate of the positive DVT cases is relatively low. The reported prevalence of DVT in our study corresponds to the data from (very few available) prospective studies in this field. The number of the DVT cases  (low event rate) is the limitation of our study which was mentioned in the discussion regarding the study limitations. Among available statistical tests that we analysed (most of them with limited prediction value), we decided to use the logistic regression analysis, and we are aware that the power of inference would probably be greater with a higher number of patients and the incidence of thrombosis. The dichotomous (binary) distribution of the dependent variable (thrombosis present or not) additionally determined that the logistic regression model was chosen.

2.I found that the study objectives are ambiguous in this study. Did you intended only to assess the frequency of DVT after varicose vein surgery? VTE risk score such as Caprini score is established to evaluate the incidence of DVT after surgery. Do varicose vein surgery have extra higher risk of DVT than other surgery? Clinical implication of this study is unclear.

Answer:

First of the aims of the study was to evaluate the frequency of the postoperative DVT in the patients undergoing varicose vein surgery (striping and miniphlebectomy) under the spinal anesthesia in the group of patients receiving routine thromprophylaxis. The second aim was to assess the utility of the Caprini score as a tool which identified the patients with the highest risk of postoperative DVT. There is no validated model of VTE risk yet, as well as no validated VTE prophylaxis protocol dedicated to the patients undergoing varicose vein surgery available. The studies which validated Caprini score concerned general surgery, intensive care, plastic surgery and also some other specialities.  but we do not know whether this score can be used in vascular surgery as well as varicose vein surgery.  In some phlebology and surgery centers, Caprini score is widely and routinely used (extrapolating the experience from general surgery studies), but in reality, no validated and efficient model of VTE risk assessment in this particular population is known. An introduction of the Caprini score evaluation as the part of the study should answer the question: If this particular VTE risk evaluation model is effective in the identification of the patients with highest risk of DVT development or perhaps another model of the risk assessment should be proposed?  

Round 2

Reviewer 1 Report

The authors gave acceptable answers to the questions and the changes were made to the text. I have no further comment to make.

Reviewer 2 Report

I thank the authors for addressing several of my remarks and the fact that they made an effort to revise the manuscript. However, several points still require attention, all of which had also been highlighted in my previous review. Most importantly, the methods used are still insufficient for external validation of a risk assessment model (see below).

Methods:

  • Study design (cohort, case series, etc) should be explicated
  • N of excluded patients should be added
  • Which variables were included for screening in the univariable assessment should be clearly stated
  • CEAP classification should be added to method, not results
  • Methods are still not suitable for external validation of a risk assessment model. Also see this paper: Assessing the Performance of Prediction Models: A Framework for Traditional and Novel Measures doi:10.1097/EDE.0b013e3181c30fb2. Only adding a single C-statistic does not suffice, at least an assessment of model calibration is needed. On the other hand, there are probably too few outcome events for strong external validation , meaning the authors should strongly consider revising the manuscript to emphasize, throughout, the descriptive and explorative nature of these data. In the current version of the manuscript it is suggested the Caprini score was not useful but in my view this cannot be inferred from the data (too limited). Also, the limitations and especially the consequences of these limitations are still only very briefly highlighted.

Results:

  • Please add to the manuscript your statement that no patient was lost to follow-up.
  • Figure 2 does not display a distribution of data in contrary to the authors statements. It shows individual data points ranked from lowest to highest (and is quite uninformative).

I still do not understand the added value of the correlations. Any weak correlation may be statistically significant however may also exist solely on coincidence. Since you found no link between D-dimer and VTE risk in the models I struggle to see the relevance of these correlations. They only distract from the other messages of your manuscript

Reviewer 3 Report

Authors described my concerns and the manuscript has been improved.